# First Detection and Genome Characterization of a New RNA Virus, Hibiscus Betacarmovirus, and a New DNA Virus, Hibiscus Soymovirus, Naturally Infecting *Hibiscus* spp. in Hawaii

**DOI:** 10.3390/v15010090

**Published:** 2022-12-29

**Authors:** Xupeng Wang, Adriana E. Larrea-Sarmiento, Alejandro Olmedo-Velarde, Alexandra Kong, Wayne Borth, Jon Y Suzuki, Marisa M Wall, Michael J Melzer, John Hu

**Affiliations:** 1Department of Plant and Environmental Protection Sciences, University of Hawaii, Honolulu, HI 96822, USA; 2United States Department of Agriculture Agricultural Research Service, U.S. Pacific Basin Agricultural Research Center, Hilo, HI 96720, USA

**Keywords:** hibiscus, high-throughput sequencing, betacarmovirus, soymovirus

## Abstract

Hibiscus (*Hibiscus* spp., family Malvaceae) leaves exhibiting symptoms of mosaic, ringspot, and chlorotic spots were collected in 2020 on Oahu, HI. High-throughput sequencing analysis was conducted on ribosomal RNA-depleted composite RNA samples extracted from symptomatic leaves. About 77 million paired-end reads and 161,970 contigs were generated after quality control, trimming, and de novo assembly. Contig annotation with BLASTX/BLASTN searches revealed a sequence (contig 1) resembling the RNA virus, hibiscus chlorotic ringspot virus (genus *Betacarmovirus*), and one (contig 2) resembling the DNA virus, peanut chlorotic streak virus (genus *Soymovirus*). Further bioinformatic analyses of the complete viral genome sequences indicated that these viruses, with proposed names of hibiscus betacarmovirus and hibiscus soymovirus, putatively represent new species in the genera *Betacarmovirus* and *Soymovirus*, respectively. RT-PCR using specific primers, designed based on the retrieved contigs, coupled with Sanger sequencing, further confirmed the presence of these viruses. An additional 54 hibiscus leaf samples from other locations on Oahu were examined to determine the incidence and distribution of these viruses.

## 1. Introduction

The flowering shrub, hibiscus (*Hibiscus* spp., family Malvaceae, order Malvales) is cultivated as an ornamental plant in Hawaiian landscapes. Numerous plant viruses have been reported infecting this plant, including virus members of the genera *Tobamovirus, Betacarmovirus, Begomovirus, Cilevirus, Higrevirus*, and *Ilarvirus*. Infected plants exhibit symptoms of mosaic, ringspot, and chlorotic spots [1]. Viruses reported infecting hibiscus in Hawaii include a cilevirus, a higrevirus, a carmovirus, and a tobamovirus [2,3].

High-throughput sequencing (HTS) is a powerful tool for virus detection and discovery, and has been used extensively for this purpose in the last decade [4]. In the spring of 2020, hibiscus plants with suspected viral symptoms were observed on the University of Hawaii at Manoa campus, but the symptoms were somewhat different from those reported previously in Hawaii. Samples from these plants were used for further study using HTS. 

## 2. Materials and Methods

### 2.1. Plant Material Collection and Sample Preparation 

*Hibiscus rosa-sinensis* with foliar mosaic, ringspot, and chlorotic spots (Figure 1) were observed on the University of Hawaii at Manoa campus (21°18′07′′ N, 157°48′55′′ W), Oahu, Hawaii, in April 2020. In total, 3 samples were collected from the campus for HTS analysis and 54 from a virus survey of other locations on Oahu for a virus survey. Total RNAs were extracted from the symptomatic leaves using a Spectrum^®^ Plant Total RNA Kit (Sigma-Aldrich, St. Louis, MO, USA) according to the manufacturer’s protocol. Quantification and quality controls of total RNAs were performed using the NanoDrop 2000/2000c spectrophotometer (Thermo Fisher Scientific Inc., Waltham, MA, USA). Three of the total RNA samples were equimolarly pooled to obtain composite samples. Ribosomal RNAs (rRNAs) were removed from the composite samples using RiboMinus^TM^ Plant Kit (Thermo Fisher Scientific Inc.)and the rRNA-depleted RNA samples used to prepare a cDNA library using the Illumina TruSeq^®^ RNA Sample Prep Kit (Illumina, San Diego, CA, USA). 

### 2.2. HTS Sequencing and Genome Assembly 

HTS was conducted on the cDNA libraries using an Illumina NextSeq 500 sequencer generating 75-bp paired-end reads. Raw reads were trimmed to remove low-quality and adapter sequences using Trimmomatic^®^ (Galaxy Version 0.38.0) and subsequently used with Trinity^®^ (Galaxy Version 2.9.1+galaxy2) for de novo assembly into contigs. Contigs were then queried with BLASTx/BLASTn to identify sequences that matched reference sequences of viruses and viroids (http://www.ncbi.nlm.nih.gov/genome/viruses/, accessed on 25 October 2022). The 5′- and 3′-end sequences of the RNA viral genomes were obtained by rapid amplification of cDNA ends (RACE) using a SMARTer^®^ RACE 5′/3′ Kit (Takara, San Jose, CA, USA) following the manufacturer’s instructions. All amplicons were either directly sequenced or cloned into a pRACE vector (Promega, Madison, WI, USA) and at least three independent clones were sequenced. Complete viral sequences from HTS and RACE with similarity to betacarmoviruses and soymoviruses were then used as reference sequences using the Bowtie2 mapper (Version 2.4.5) plug-in implemented in Geneious^®^ Prime (Version 2022.2.1) and raw reads. 

### 2.3. RT-PCR Detection and Virus Survey

Contig-specific primer sequences were designed with Primer3 and BLASTn and used in RT-PCR assays to confirm the presence of individual viruses and for HTS validation (Table 1). Total RNAs were reverse transcribed into cDNAs using random primers and the M-MLV reverse transcription kit (Promega, WI, USA). GoTaq^®^ Green Master Mix (Promega) was used in all PCR reactions. Cycling conditions for virus-specific primer sets were as follows: a single cycle at 95 °C for 5 min; followed by 35 cycles at 95 °C for 30 s, 55 °C for 30 s, and 72 °C for 45 s; a single cycle at 72 °C for 5 min; and then held at 4 °C until analysis. RT-PCR products were resolved on 1.5% agarose gels stained with ethidium bromide and visualized under UV light. Amplicons were either sequenced directly, or first ligated into a linearized pRACE vector (Takara). Three individual clones of each PCR product with inserts of the expected sizes were then selected and sequenced in both directions by Sanger sequencing (Genewiz, San Diego, CA, USA). An additional 54 symptomatic and asymptomatic leaves from various *Hibiscus* spp. were sampled from other locations on Oahu. 

### 2.4. Phylogenetic and Bioinformatic Analyses

For phylogenetic analyses, whole genome sequences from homologs of relevant reference viruses were obtained from GenBank and MUSCLE aligned with the genome sequences generated in this study using Geneious^®^ Prime (Version 2022.2.1). A phylogenetic tree was constructed in MEGA X [5] using a maximum likelihood algorithm with a K2 + G + I substitution model. One thousand bootstrap repetitions provided statistical support for groups within the trees. Geneious^®^ Prime (Version 2020.2.4) was also used for bioinformatic analyses of the genome sequences, including identification of ORFs, prediction of encoded proteins, pairwise sequence comparisons of nucleotide and protein amino acids, and sequence alignments.

### 2.5. Mechanical Inoculation

Sixteen plants of *Nicotiana benthamiana* and five plants of *N. tabacum* were mechanically inoculated using carborundum and extracts obtained from hibiscus plants infected with hibiscus betacarmovirus and hibiscus soymovirus. Three plants were mock-inoculated. The extracts were prepared by homogenizing plant leaf tissue with sodium phosphate buffer (20 mM sodium phosphate, pH 7.0) at a ratio of 1:10 (*w/v*). 

## 3. Results 

### 3.1. HTS Data Analyses 

About 77.4 M paired-end reads were generated and 162,000 contigs obtained after quality control, trimming, and de novo assembly, as detailed in Section 2.2. Contig annotation with BLASTX/BLASTN against the viral database in GenBank revealed the presence of two viruses, a betacarmovirus that we called hibiscus betacarmovirus (HBCV) and a soymovirus that we called hibiscus soymovirus (HSV). Of ~76.8 M HTS trimmed reads, 8,721,585 (11.35%) and 7717 (0.01%) reads were mapped onto the complete genome sequences of hibiscus HBCV and HSV, respectively. The three samples used for HTS were also subjected to individual RT-PCR detection and tested positive to both viruses. 

### 3.2. Identification of a New RNA Virus, Hibiscus Betacarmovirus (HBCV)

The complete genome sequence of HBCV obtained by HTS and RACE consisted of 3815 nucleotides (GenBank accession number OP757658). HBCV shared the highest whole genome nucleotide sequence identity (68.4%) with a hibiscus chlorotic ringspot virus (HCRSV) Israel isolate (KC876666), and the highest whole genome amino acid sequence identity (58.8%) with a HCRSV South Africa isolate (UUU45889). Three putative ORFs were found in the HBCV genome. HBCV ORF1 (nt 36–2276) encoded a putative RNA-dependent RNA polymerase (RdRp) of 747 aa, ORF2 (nt 2243–2464) encoded a putative viral movement protein of 222 aa, and ORF3 (nt 2610–3677) encoded a putative viral coat protein (CP) of 356 aa. 

### 3.3. Identification of a New DNA Virus, Hibiscus Soymovirus (HSV)

The complete genome sequence of hibiscus soymovirus (HSV) obtained by HTS consisted of 8143 nucleotides (GenBank accession number OP757659). Ten putative ORFs and their locations were found in the HSV genome: ORF1 (nt 45–296), ORF2 (nt 372–800), ORF3 (nt 797–1129), ORF4 (nt 1131–1298), ORF5 (nt 1298–2710), ORF6 (nt 2703–4760), ORF7 (nt 4739–5005), ORF8 (nt 5319–6125), ORF9 (nt 6769–7197), and ORF10 (nt 7190–8095), respectively. HSV ORF6 encoded a putative viral replicase with the conserved domains of reverse transcriptase (RT) and ribonuclease H (RNase H). 

### 3.4. Virus Survey

Additional hibiscus leaf samples from other locations were examined to determine the incidence, distribution, and diversity of these viruses in Hawaii. Twelve of the fifty four samples (22%) tested were positive for at least one of the viruses: One sample tested positive for HBCV, nine samples tested positive for HSV, and two samples tested positive for both HBCV and HSV. 

### 3.5. Mechanical Inoculation

We tried to passage the two viruses from hibiscus plants to 16 *N. benthamiana* and 5 *N. tabacum* plants, by mechanical inoculation. To confirm viral infection, the newest fully expanded leaf of each plant at six weeks post-inoculation was tested by RT-PCR with the specific primers as described above. The RNA used for HTS was used as a positive control for RT-PCR. RT-PCR results showed that the positive control had band with the expected size, while the no template control and the RNA samples from these inoculated plants did not have band with the expected size. The mechanical inoculation experiments and RT-PCR assays suggested that HBCV and HSV could not be mechanically inoculated into either *N. benthamiana* or *N. tabacum* using the present protocol.

## 4. Discussion

The criteria for defining a species in the genus *Betacarmovirus* are RdRp amino acid sequence identity of less than 57% and CP amino acid sequence identity of less than 52% [2]. The amino acid sequence of the RdRp of HBCV was closest in identity (53.3%) to that of the RdRp of the HCRSV Israel isolate while the amino acid sequence of the CP of HBCV was closest in identity (34.0%) to that of the CP of the HCRSV Singapore isolate (Table 2). These results suggested that HBCV belongs to a new virus species, among carmoviruses. *Carmovirus*, once the largest genus in the family *Tombusviridae*, has been sub-divided into three new genera, *Alphacarmovirus, Betacarmovirus,* and *Gammacarmovirus*, because of the complexity of its phylogenetic tree [6,7]. The genomic nucleotide sequences of HBCV and members of these three genera were used to construct a phylogenetic tree. The phylogenetic analysis placed HBCV in the *Betacarmovirus* clade closely related to HCRSV (Figure 2). Thus, we propose to assign HBCV to the *Betacarmovirus HBCV* species.

The criterion demarcating species in the genus *Soymovirus* is a difference in polymerase (RT + RNase H) nt sequences of more than 20% [8]. The nucleotide sequence of the HSV polymerase had the closest identity (71.1%) to that of the polymerase to the soymovirus, peanut chlorotic streak caulimovirus (U13988). These results suggest that HSV belongs to a new virus species that we propose to appoint *Soymovirus HSV,* in the genus *Soymovirus*. Soymoviruses are single-component double-stranded DNA viruses with only four member species at present (https://ictv.global/report/chapter/caulimoviridae/caulimoviridae/soymovirus, accessed on 25 October 2022) and two putative additional species [9,10]. The phylogenetic analysis shows that HSV is closest to peanut chlorotic streak virus (Figure 3). 

In this study, we detected two new viruses infecting hibiscus with symptoms that were different from those previously reported in Hawaii. This may be a new plant virus disease that has appeared on hibiscus in Hawaii. 

## Figures and Tables

**Figure 1 viruses-15-00090-f001:**
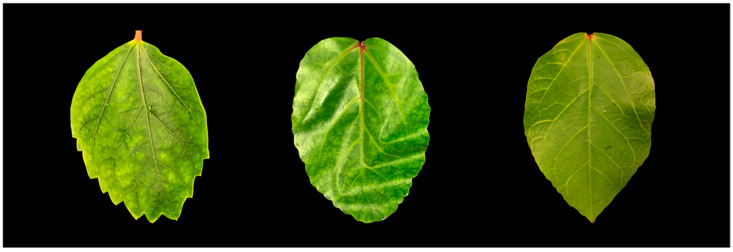
Symptomatic hibiscus leaves (**left** and **middle**) and a healthy leaf (**right**) were collected on the island of Oahu, Hawaii.

**Figure 2 viruses-15-00090-f002:**
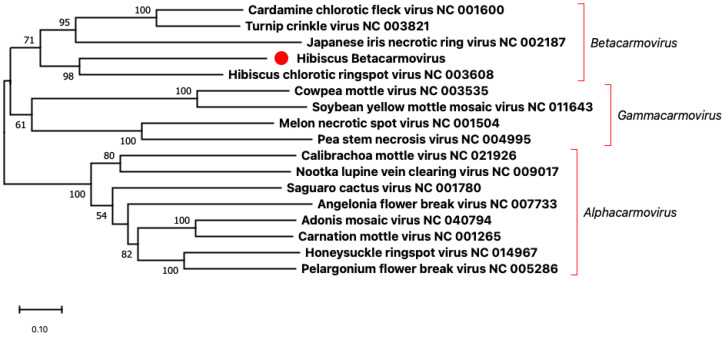
Phylogenetic analysis based on the genomic nucleotide sequences of members of the genera *Alphacarmovirus*, *Betacarmovirus*, and *Gammacarmovirus* was performed using the Maximum Likelihood method and MEGA X software with 1000 bootstrap repetitions. The red dot indicates the new virus hibiscus betacarmovirus that we found in this study.

**Figure 3 viruses-15-00090-f003:**
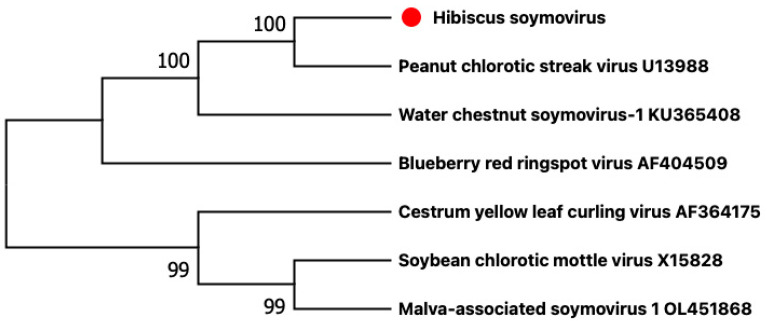
Phylogenetic analysis based on the genomic nucleotide sequences of members of the genus *Soymovirus* was performed using the maximum likelihood method and MEGA X software with 1000 bootstrap repetitions. The red dot indicates the new virus hibiscus soymovirus that we found in this study.

**Table 1 viruses-15-00090-t001:** Primers used in the present study for RACE experiments and detection of hibiscus betacarmovirus (HBCV) and hibiscus soymovirus (HSV) using RT-PCR assays.

Purpose	ID	Sequence-5′ to 3′	Length
HBCV detection	BCVF1	CCTGAGGTTTGAGCACAGCA	659 bp
	BCVR1	GACCAAGGCTCCTCTTTGCA	
HSV detection	SVF1	AGGAAGATGGTCGTTTTGGG	429 bp
	SVR1	ACTGGACTGCTGGTTGTATG	
5′-RACE	HBCV _GSP1	GATTACGCCAAGCTTTGGGTTATGCCCAAGACCAC	-
3′-RACE	HBCV _GSP2	GATTACGCCAAGCTTAAAACCGGTGTTATTGCCGC	-

**Table 2 viruses-15-00090-t002:** Comparison of amino acid sequences of RNA-dependent RNA polymerase (RdRp) and coat protein (CP) from different hibiscus chlorotic ringspot virus isolates with those of hibiscus betacarmovirus.

Isolate	Nucleotide	Amino Acid	RdRp	CP
Israel	KC876666	AGN70380	53.3%	32.6%
Taiwan	DQ392986	ABD48708	52.9%	33.1%
Malaysia	MN080500	QDJ95885	53.3%	33.4%
Singapore	X86448	CAB81767	53.0%	34.0%
Hawaii	MT512573	QWX21617	53.0%	33.7%
China	KY933060	AVK94112	53.1%	34.0%

## Data Availability

The genomic sequences of hibiscus betacarmovirus (HBCV) and hibiscus soymovirus (HSV) were submitted to GenBank (GenBank accession number OP757658 and OP757659).

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
