# Peer review of "First Detection and Genome Characterization of a New RNA Virus, Hibiscus Betacarmovirus, and a New DNA Virus, Hibiscus Soymovirus, Naturally Infecting Hibiscus spp. in Hawaii"

_viruses, 2022, doi:10.3390/v15010090_

Round 1

Reviewer 1 Report

In present study, the authors survey the viruses infecting Hibiscus spp. leaves exhibiting symptoms of mosaic, ringspot, and chlorotic spots on Oahu, HI. Two possible novel plant viruses were found based on high-throughput sequencing, RACE and Sanger sequencing. The authors also survey the incidence and distribution of these two viruses. Overall, the data of results is solid, the manuscript is well organized and presented. However, several points should be addressed before the MS accepted for publication in Viruses:

1)        Line 14, “Hibiscus sp.” should be “Hibiscus spp.”.

2)        In present, numerous plant viruses have been reported infecting this plant, including virus members of the genera Tobamovirus, Betacarmovirus, Begomovirus, Cilevirus, Higrevirus, and Ilarvirus. In this study, how about the incidence of previously known viruses in Hibiscus spp..   

3)        Only Nicotiana benthamiana and N. tabacum were selected and inoculated, how about other indicator hosts and natural host (Hibiscus spp.)?

4)        Line 132, “Table 2” should be “Table 3”.

5)        Only limited information are included in Table 3. I think the result of virus survey just show in the text.

6)        Many editorial errors like above were also found in the MS and references. The authors should rewrite this MS carefully.

Author Response

In present study, the authors survey the viruses infecting Hibiscus spp. leaves exhibiting symptoms of mosaic, ringspot, and chlorotic spots on Oahu, HI. Two possible novel plant viruses were found based on high-throughput sequencing, RACE and Sanger sequencing. The authors also survey the incidence and distribution of these two viruses. Overall, the data of results is solid, the manuscript is well organized and presented. However, several points should be addressed before the MS accepted for publication in Viruses:

1) Line 14, “Hibiscus sp.” should be “Hibiscus spp.”.

R: Change implemented

2) In present, numerous plant viruses have been reported infecting this plant, including virus members of the genera Tobamovirus, Betacarmovirus, Begomovirus, Cilevirus, Higrevirus, and Ilarvirus. In this study, how about the incidence of previously known viruses in Hibiscus spp..

R: Viruses reported infecting hibiscus in Hawaii include a cilevirus (citrus leprosis virus cytoplasmic type 2), a higrevirus(hibiscus yellow blotch virus), a carmovirus (hibiscus chlorotic ringspot virus), and a tobamovirus (hibiscus latent Fort Pierce virus). In this study, we detected thehibiscus latent Fort Pierce virus, but not the other three viruses.

3) Only Nicotiana benthamiana and N. tabacum were selected and inoculated, how about other indicator hosts and natural host(Hibiscus spp.)?

R: Larger host range testing experiments are still in progress. Our main intention in this short Communication is to publish the sequence information as soon as possible.

4) Line 132, “Table 2” should be “Table 3”.

R: We took the following suggestion and deleted Table 3.

5) Only limited information are included in Table 3. I think the result of virus survey just show in the text.

R: Change implemented.

6) Many editorial errors like above were also found in the MS and references. The authors should rewrite this MS carefully.

R: Thank you for your suggestion, we double-checked and revised the manuscript.

Reviewer 2 Report

In this communication, the authors describe two new viruses they identified by high throughput sequencing of samples collected from symptomatic hibiscus at the campus on the Ohau island in Hawaï. One is a (+) sense RNA virus and the second one is a dsRNA virus. Based on ICTV species demarcation criteria, it is suggested that they both belong to new species, in the genera Betacarmovirus and Soymovirus, respectively.

Main concerns:

There seem to be some confusion between species and virus; what is present in a plant and was sequenced is a virus belonging to a species. Authors can describe and name new viruses but only can propose new species for ICTV approval.

The authors describe how they obtained sequence data and assembled contigs (lines 106-107). They then give percentage of reads mapping to "Hibiscus betacarmovirus (HBCV) and Hibiscus soymovirus (HSV), respectively". It is not clear to what the authors gave these names? contigs? viruses? virus species? It is not clear whether these are new species that should be written in italics with the first letter in capital. Remember that new species and species names can only be proposed as they need to be approved by ICTV. Or are these virus names that should not have a capital letter as hibiscus is also a common name (not only a genus name)? If these are viruses they can indeed be abbreviated as HBCV and HSV.

I suggest the authors present the sequence results, give a name for the viruses and an abbreviation, present the percentage identity with other viruses of the genus, as part of the results, and propose a species name in accordance to ICTV rule "The first word component shall begin with a capital letter and be identical in spelling to the name of the genus to which the species belongs. The second word component shall not contain any suffixes specific for taxa of higher ranks. The entire species name (both word components) shall be italicized."

From the abstract I understand that the 54 samples used for the survey originate from different locations of the Ohau island (lines 25-26) whereas from the mat & meth part I understand that they originate from the same location (campus) as the three samples used for the sequencing. Please clarify.

As the paper is aimed to be published in a special issue on emerging plant viruses, could this concept be discussed for the 2 new viruses?

Additional comments:

Lines 23, 85: bad hyphenations

Lines 22, 114, 122: remove capital letter from the word hibiscus which is a common name in English, not only a host genus.

Line 44: refer to Fig1 when describing symptoms.

Lines 51-52: how were the rRNAs depleted, was it by selecting polyA RNAs?

Line 105: previously is somehow confusing; if it refers to a previous publication, the publication should be quoted and if it refers to the mat & meth part of the current paper then previously should be replaced with "in section 2.2".

Line 135: specify "could not be inoculated using our/the present protocol" as this does not mean that the viruses are not mechanically inoculable.

Line 137: correct genus Betacarmovirus.

Line 152: correct genus Soymovirus.

Line 155: correct peanut chlorotic streak virus.

Table 3 should be completed with the presence/absence of symptoms. Also prefer minus sign to hyphen to represent negative RT-PCR. Were the 3 samples used for sequencing also subjected to individual RT-PCR detection?

References:

- remove extra capital letters in titles.

- check doi of ref 1 as it is the one of ref 3.

- use italics where appropriate e. g. Hibiscus rosa-sinensis and Eleocharis dulcis.

- complete initials, e. g. ref 4: Villamor D.E.V or ref 7.

Add ratification for the split of the 3 carmovirus genera: Adams et al., Arch Virol (2016) 161:2921–2949, DOI 10.1007/s00705-016-2977-6

- check the order of authors, e. g. ref 7.

Round 2
